# Integrative Taxonomy of Novel *Diaporthe* Species Associated with Medicinal Plants in Thailand

**DOI:** 10.3390/jof9060603

**Published:** 2023-05-24

**Authors:** Jutamart Monkai, Sinang Hongsanan, Darbhe J. Bhat, Turki M. Dawoud, Saisamorn Lumyong

**Affiliations:** 1Research Center of Microbial Diversity and Sustainable Utilization, Faculty of Science, Chiang Mai University, Chiang Mai 50200, Thailand; mjutamart@gmail.com (J.M.); sinang333@gmail.com (S.H.); 2Department of Biology, Faculty of Science, Chiang Mai University, Chiang Mai 50200, Thailand; 3Department of Botany & Microbiology, College of Science, King Saud University, P.O. Box 2455, Riyadh 11451, Saudi Arabia; bhatdj@gmail.com (D.J.B.); tdawoud@ksu.edu.sa (T.M.D.); 4Vishnugupta Vishwavidyapeetam, Ashoke, Gokarna 581326, India; 5Academy of Science, The Royal Society of Thailand, Bangkok 10300, Thailand

**Keywords:** Asexual morph, classification, Diaporthaceae, new host records, saprobes

## Abstract

During our investigations of the microfungi on medicinal plants in Thailand, five isolates of *Diaporthe* were obtained. These isolates were identified and described using a multiproxy approach, viz. morphology, cultural characteristics, host association, the multiloci phylogeny of ITS, *tef1-α*, *tub2*, *cal*, and *his3*, and DNA comparisons. Five new species, *Diaporthe afzeliae*, *D. bombacis*, *D. careyae*, *D. globoostiolata*, and *D. samaneae*, are introduced as saprobes from the plant hosts, viz. *Afzelia xylocarpa*, *Bombax ceiba*, *Careya sphaerica*, a member of Fagaceae, and *Samanea saman*. Interestingly, this is the first report of *Diaporthe* species on these plants, except on the Fagaceae member. The morphological comparison, updated molecular phylogeny, and pairwise homoplasy index (PHI) analysis strongly support the establishment of novel species. Our phylogeny also revealed the close relationship between *D. zhaoqingensis* and *D. chiangmaiensis*; however, the evidence from the PHI test and DNA comparison indicated that they are distinct species. These findings improve the existing knowledge of taxonomy and host diversity of *Diaporthe* species as well as highlight the untapped potential of these medicinal plants for searching for new fungi.

## 1. Introduction

Medicinal plants are essential for sustaining human health and livelihoods according to their ethnobotanical uses and therapeutic purposes [1,2]. They have also contributed to maintaining biodiversity in forest ecosystems and supporting natural recreation in urban ecosystems [1,2]. Fungi are usually encountered in medicinal plants, where they can affect their hosts in both beneficial and harmful manners [2,3,4]. As pathogens, they impair plant health and productivity [4]; whereas, as endophytes, they promote plant growth and produce a diverse array of secondary metabolites, which have been exploited for the development of new drugs and pharmaceutical products [2,3]. Thus, studies of fungi associated with medicinal plants represent a significant repository for the estimation of fungal diversity, the discovery of novel fungi and fungal–plant interactions, as well as the bioprospecting of new bioactive compounds and their biotechnological applications [5,6,7,8,9,10,11,12].

*Diaporthe* species are a large and diverse group of fungi known as endophytes, saprobes, and plant pathogens, with worldwide distribution and a broad range of host associations [13,14,15,16,17,18]. Pathogenic *Diaporthe* species cause various plant diseases, such as blight, cankers, diebacks, fruit rots, leaf spots, and wilts, on forest trees [19,20,21,22] and many agricultural crops such as citrus, grapevine, peach, soybean, sunflower, and tea [23,24,25,26,27,28]. Morphologically, *Diaporthe* is characterized by pseudostromatic ascomata that usually have black lines in the host substrate, along with elongated perithecial necks for the sexual morph [29], and asexual morph consisting of ostiolate conidiomata, aseptate, and polymorphic (alpha, gamma, and beta), and hyaline conidia [14]. However, identifying *Diaporthe* species based solely on morphological data is challenging due to their polyphyletic nature and the presence of numerous cryptic species [30,31,32]. Recent studies have used multilocus phylogeny, including internal transcribed spacers (ITS), the translation elongation factor 1-α (*tef1-α*), β-tubulin (*tub2*), calmodulin (*cal*), and histone H3 (*his3*), along with morphological characteristics, to accurately identify and classify *Diaporthe* species [15,19,23,26,31,33,34,35,36]. Norphanphoun et al. [32] classified *Diaporthe* into 13 species complexes based on a comprehensive sequence dataset of five loci (ITS, *tef1-α*, *tub2*, *cal*, and *his3*) to assist species delineation. The integrative approach based on cultural, ecological, morphological, and molecular characteristics is advantageous for accurately identifying *Diaporthe* species [22,27,28,35,36,37].

Taxonomic studies of *Diaporthe* revealed a variety of medicinal plants as their hosts [38]. However, most of these studies have been conducted in temperate zones (i.e., [15,16,17,21,24,26,28]). Knowledge of *Diaporthe* associated with medicinal plants in the tropics is still limited [31,32]. Therefore, this study aims to identify and describe isolates of *Diaporthe* associated with several medicinal plants in Thailand using both morphological and molecular analyses. To better illustrate the placements of the five new species, their morphological descriptions, micrographs, and updated phylogenetic trees are presented and discussed.

## 2. Materials and Methods

### 2.1. Sample Collection and Morphological Examination

Fresh fungal specimens were collected from the dead leaves and woody twigs of various medicinal plants in urban parks and forest areas in the Chiang Mai and Tak provinces of Thailand in 2019 and 2022. Collected samples were investigated for macro- and micro-morphological structures using a Nikon SMZ800N stereo microscope (Nikon Instruments Inc., Melville, NY, USA) and photomicrographed with a Nikon Eclipse Ni compound microscope attached to a Nikon DS-Ri2 camera system (Nikon Instruments Inc., Melville, NY, USA). The measurement of each structure (i.e., conidiomata, conidiomatal walls, conidiophores, conidiogenous cells, and conidia) was taken using the Tarosoft (R) Image Frame Work program. All figures were modified using Adobe Photoshop CS6 Extended version 10.0 software (Adobe Systems, San Jose, CA, USA).

### 2.2. Fungal Isolation and Preservation

Pure cultures were obtained from single spore isolation on 2% water agar (WA), and germinated conidia were aseptically transferred to potato dextrose agar (PDA) [39]. Fungal cultures were incubated at 25 °C for four to six weeks and then examined for colony morphology and spore production. Herbarium material and pure culture of *Diaporthe globoostiolata* were deposited in the herbarium of Mae Fah Luang University (MFLU) and the Mae Fah Luang University Culture Collection (MFLUCC), Chiang Rai Province, Thailand. Herbarium materials and pure cultures of *D. afzeliae*, *D. bombacis*, *D. careyae*, and *D. samaneae* were deposited in the Herbarium of the Department of Biology (CMUB) and the Culture Collection of Sustainable Development of Biological Resources Laboratory, Faculty of Science, Chiang Mai University (SDBR-CMU), Chiang Mai Province, Thailand. The numbers of Index Fungorum and Faces of Fungi were acquired as outlined in the Index Fungorum [40] and Jayasiri et al. [41].

### 2.3. DNA Extraction, PCR Amplification, and Sequencing

A DNA Extraction Mini Kit (FAVORGEN, Ping-Tung, Taiwan) was used to extract genomic DNA from fungal colonies grown on PDA for two weeks. Five phylogenetic markers including internal transcribed spacers (ITS), translation elongation factor 1-α (*tef1-α*), β-tubulin (*tub2*), calmodulin (*cal*), and histone H3 (*his3*) were amplified using the primer pairs ITS5/ITS4 [42], EF1-728F/EF1-986R [43], Bt2a/Bt2b [44], CAL228F/CAL737R [43], and CYLH3F/H3-1b [44,45], respectively. The PCR conditions for each gene region were carried out as described by Jiang et al. [21]. The purification of PCR products was processed using a PCR Clean-up Gel Extraction NucleoSpin^®^ Gel and PCR Clean-up Kit (Macherey-Nagel, Düren, Germany). The sequence analysis was operated by the genetic analyzer at 1^ST^ Base Company (Kembangan, Malaysia).

### 2.4. Phylogenetic Analyses

The sequences obtained in this study were submitted through a BLASTn search in GenBank (www.ncbi.nlm.nih.gov/blast/, assessed on 1 March 2023) to determine the most similar taxa. The initial phylogenetic analysis was conducted based on the ITS sequence dataset from Norphanphoun et al. [32] to identify the placement of our isolates within species complexes. The newly generated sequences and their related sequences were then selected for the concatenated ITS, *tef1-α*, *tub2*, *cal*, and *his3* sequence dataset based on the BLASTn search results and updated literature [18,22,32,46,47,48] (Table 1). The alignment of a single locus dataset was performed using MAFFT v.7 (http://mafft.cbrc.jp/alignment/server/index.html, assessed on 1 March 2023) [49] and the ambiguous sites were manually adjusted using BioEdit 7.1.3.0 [50]. The phylogenetic trees of single locus and combined datasets were analyzed using maximum likelihood (ML) and Bayesian inference (BI) criteria. Tree topologies from single locus analyses were also compared and no conflicts were found.

ML and BI analyses were performed using RAxML-HPC2 on XSEDE (v.8.2.12) [51] and MrBayes on XSEDE v.3.2.7a [52,53,54] in the CIPRES Science Platform V3.3 (https://www.phylo.org/portal2/home.action, assessed on 1 March 2023) [55]. The GTRGAMMA model of the bootstrapping phase with 1000 bootstrap iterations was set as the parameter for ML analysis [51]. The best nucleotide substitution model was determined using MrModeltest v.2.3 [56], and GTR + I + G was selected as the best-fitting model for the ITS, *tef1-α*, *tub2*, *cal*, and *his3* datasets. For BI analysis, six simultaneous Markov chains were set to run 10,000,000 generations with a sampling frequency of 100 generations. The burn-in phase was set as 0.25, and the posterior probabilities (PP) were evaluated from the remaining trees. The phylogenetic trees resulting from the ML and BI analyses were visualized by the FigTree v1.4.0 program [57] and adjusted using Adobe Photoshop CS6 software (Adobe Systems, San Jose, CA, the USA). Novel obtained sequences were registered for GenBank accession numbers.

### 2.5. Genealogical Concordance Phylogenetic Species Recognition Analysis

The recombination level between new species and their most closely related taxa was examined using the Genealogical Concordance Phylogenetic Species Recognition (GCPSR) model [58,59]. A pairwise homoplasy index (PHI) test was implemented by SplitsTree4 using the LogDet transformation and split decomposition options [60,61]. A PHI test result (Φw) above 0.05 indicated no significant recombination in the dataset. In addition, split graphs were generated for visualization of the relationship between closely related species.

## 3. Results

### 3.1. Molecular Phylogeny

The combination of the ITS, *tef1-α*, *tub2*, *cal*, and *his3* sequence datasets comprised 191 *Diaporthe* strains, with *Cytospora disciformis* CBS 116,827 and *C. leucostoma* SXYLt as the outgroups. The aligned sequence dataset contained a total of 3020 characters with gaps in the order of ITS (1–588), *tef1-α* (589–992), *tub2* (993–1800), *cal* (1801–2522), and *his3* (2523–3020). The final RAxML analysis resulted in the best scoring tree with a final optimization likelihood value of -42610.603037. The matrix comprised 2000 distinct alignment patterns, with 33.63% undetermined characters or gaps. The estimated base frequencies were as follows: A = 0.218896, C = 0.324616, G = 0.235283, and T = 0.221205; substitution rates AC = 1.207023, AG = 3.073601, AT = 1.095752, CG = 0.816932, CT = 4.008593, and GT = 1.000000; and gamma distribution shape parameter of 0.398901. The phylogenetic trees generated from the ML and BI analyses revealed similar topologies. The newly recovered isolates formed five monophyletic lineages within three species complexes as follows: *D. afzeliae*, *D. bombacis*, and *D. globoostiolata* were clustered within the *D. arecae* species complex; *D. samaneae* was grouped in the *D. oncostoma* species complex; and *D. careyae* was associated with the *D. carpini* species complex (Figure 1).

### 3.2. Genealogical Concordance Phylogenetic Species Recognition Analysis

In the PHI analysis, there was no evidence of significant recombination (Φw > 0.05) between each new species (*Diaporthe afzeliae*, *D. bombacis*, *D. globoostiolata*, and *D. samaneae*) and their closely related taxa in the combined ITS, *tef1-α*, *tub2*, *cal*, and *his3* sequence dataset (Figure 2a–d). The results of PHI analysis also revealed no significant recombination (Φw > 0.05) between *D. zhaoqingensis* and *D. chiangmaiensis* (Figure 2e). This evidence confirms that they are distinct species.

### 3.3. Taxonomy

***Diaporthe afzeliae*** Monkai and *S. Lumyong*, sp. nov.

Index Fungorum number: IF900377; Faces of fungi number: FoF 14091; Figure 3

Etymology: Refers to the host genus, *Afzelia*, from which the holotype was collected.

Holotype: CMUB39998

*Saprobic* on dead wood of *Afzelia xylocarpa*. Sexual morph: undetermined. Asexual morph: Coelomycetous. *Conidiomata*: 200–300 high × 450–850 μm diam., pycnidial, stromatic, subepidermal, immersed, clustered, ovoid to subconical, elongated, dark brown to black, ostiolate, and multi-loculate. *Ostioles*: up to 120 μm wide, subglobose or conical, dark brown, and papillate. *Conidiomatal wall*: up to 40 μm wide, comprising several layers of thin-walled cells, arranged in *textura angularis*, with dark brown outer layers and hyaline to pale brown inner layers. *Conidiophores*: 9–26.7 × 1.7–3 μm (*x* = 15.8 × 2.3 μm, n = 30), tightly aggregated, subcylindrical, hyaline, septate, branched, and straight to sinuous. *Conidiogenous cells*: 8.2–18 × 1.4–2.7 μm (*x* = 12 × 2 μm, n = 30), subcylindrical to ampulliform, tapering towards apex, hyaline, phialidic, and terminal, with visible periclinal thickening and a prominent collarette. *Alpha conidi:a* 5.6–10.4 × 2.3–2.8 μm (*x* = 8.5 × 2.3 μm, n = 30), ellipsoid to elongate fusiform, obtuse at apex, subtruncate at base, sometimes with a denticle attached to the base, aseptate, hyaline, smooth-walled, and eguttulate. *Beta conidia*: not observed.

Culture characteristics: *Colonies* on PDA reached 5 cm diam. after 10 days at 25 °C, effuse, fluffy, lobate margin, originally white, becoming grey and yellow grey mycelium with age, yellowish to pale brown in reverse, with numerous black dots developing as the fruiting bodies (conidial production not seen).

Material examined: Thailand, Chiang Mai Province, Kanjanapisak Park, on dead wood of *Afzelia xylocarpa* (Kurz) Craib (Fabaceae), 4 April 2022, J. Monkai, KJ32 (CMUB39998, holotype), ex-type living culture, SDBR-CMU467.

Notes: *Diaporthe afzeliae* formed a sister clade to *D. searlei* and *D. pterocarpicola* (Figure 1). *Diaporthe afzeliae* can be distinguished from *D. searlei* CBS 146,456 by 0.84% and 2.22% base pair differences in ITS (5/598 bp) and *tef1-α* (11/495 bp) and *D. pterocarpicola* MFLUCC 10-0580 in 3.5%, 0.8%, 1.84%, and 3.79% base pair differences in ITS (18/515 bp), *tef1-α* (3/373 bp), *tub2* (8/435 bp), and *cal* (17/448 bp). *Diaporthe afzeliae* is different from *D. searlei* by its wider conidia {5.6–10.4 × 2.3–2.8 vs. 5–9 × 1.5–2 μm} [62] and *D. pterocarpicola* by its narrower conidia {5.6–10.4 × 2.3–2.8 vs. (5–)6–7(–8) × (2–)2.5(–3.5) μm} [33]. Moreover, *D. afzeliae* was isolated as a saprobe from *Afzelia xylocarpa*, while *D. searlei* was associated with the husk rot of *Macadamia* sp. [62] and *D. pterocarpicola* infected leaves of *Pterocarpus indicus* [33].

***Diaporthe bombacis*** Monkai and *S. Lumyong*, sp. nov.

Index Fungorum number: IF900378; Faces of fungi number: FoF 14092; Figure 4

*Etymology*: Refers to the host genus, *Bombax*, from which the holotype was collected.

Holotype: CMUB39995

*Saprobic* on dead wood of *Bombax ceiba*. Sexual morph: undetermined. Asexual morph: Coelomycetous. *Conidiomata*: 220–330 high × 270–430 μm diam., pycnidial, stromatic, subepidermal, immersed, clustered, subglobose to ovoid, dark brown to brown, ostiolate, and uni-to multi-loculate. *Ostioles*: up to 90 μm wide, central, subglobose, dark brown. *Conidiomatal wall*: up to 40 μm wide, comprising a few layers of thin-walled cells, arranged in *textura angularis*, with dark brown outer layers and hyaline to pale brown inner layers. *Conidiophores*: 6.2–24 × 1.5–2.8 μm (*x* = 15.8 × 2 μm, n = 30), tightly aggregated, subcylindrical, hyaline, septate, branched, and straight to sinuous. *Conidiogenous cells*: 4.5–12 × 1.4–2.4 μm (*x* = 8 × 2 μm, n = 30), subcylindrical to ampulliform, tapering towards the apex, hyaline, phialidic, and terminal, with visible periclinal thickening; collarette not observed. *Alpha conidia*: 6–9.4 × 1.7–3 μm (*x* = 7.6 × 2.4 μm, n = 30), ellipsoid to elongate fusiform, obtuse at apex, subtruncate at base, aseptate, hyaline, smooth-walled, and eguttulate. *Beta conidia*: not observed.

Culture characteristics: *Colonies* on PDA reached 5 cm diam. after 10 days at 25 °C, effuse, fluffy, lobate at the margin, originally white, becoming yellowish to pale brown mycelium with age, yellowish to pale brown in reverse, with numerous black dots developing as the fruiting bodies (conidial production not seen).

Material examined: Thailand, Chiang Mai Province, Kanjanapisak Park, on dead wood of *Bombax ceiba* L. (Bombacaceae), 4 April 2022, J. Monkai, KJ12 (CMUB39995, holotype), ex-type living culture, SDBR-CMU468.

Notes: *Diaporthe bombacis* formed a distinct clade adjacent to *D. eugeniae* (Figure 1). *Diaporthe bombacis* can be distinguished from *D. eugeniae* CBS 444.82 in 0.7%, 0.85%, 4.98%, 2.68%, and 1.92% base pair differences in ITS (4/571 bp), *tef1-α* (6/352 bp), *tub2* (20/402 bp), *cal* (13/485 bp), and *his3* (9/469 bp). *Diaporthe bombacis* resembles *D. eugeniae* in having stromatic and uni-to multi-loculate conidiomata with ostioles [63]. However, *D. bombacis* differs from *D. eugeniae* in having longer alpha conidia {6–9.4 × 1.7–3 vs. 6 × 2–3 μm} and the absence of beta conidia [63]. *Diaporthe eugeniae* was reported from *Eugenia aromatica* [63], while *D. bombacis* was found on *Bombax ceiba*.

**Figure 3 jof-09-00603-f003:**
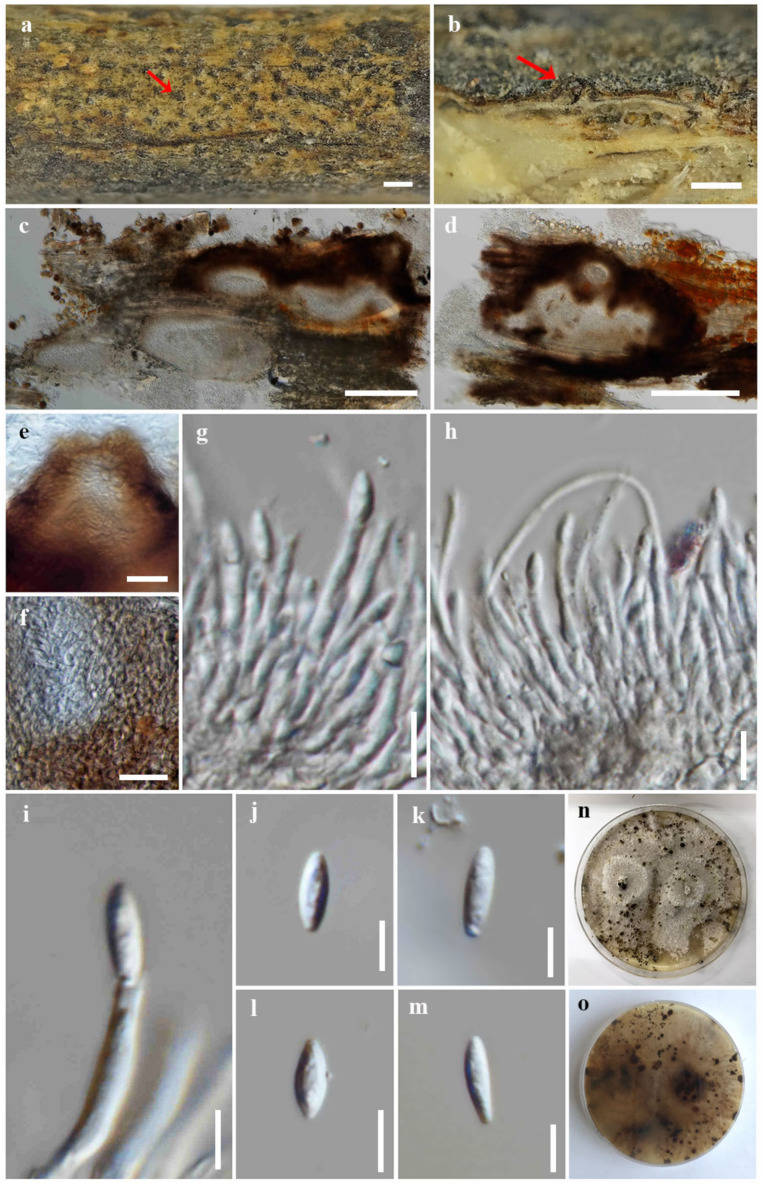
*Diaporthe afzeliae* (CMUB39998, holotype). (**a**) Conidiomata on host substrate (indicated with the red arrow). (**b**–**d**) Section through conidiomata (indicated with the red arrow). (**e**) Ostiole. (**f**) Conidiomatal walls. (**g**–**i**) Conidiogenous cells giving rise to conidia. (**j**–**m**) Alpha conidia. (**n**,**o**) Colonies on PDA, (**n**) from above and (**o**) from reverse. Scale bars: (**a**) = 500 μm, (**b**–**d**) = 200 μm, (**e**,**f**) = 20 μm, (**g**,**h**) = 10 μm, and (**i**–**m**) = 5 μm.

**Figure 4 jof-09-00603-f004:**
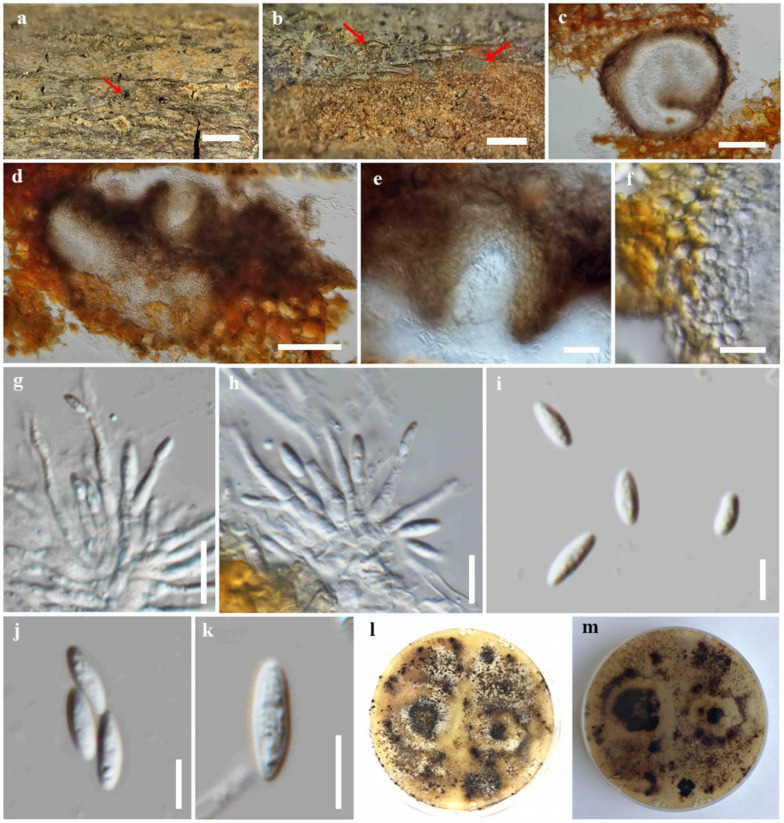
*Diaporthe bombacis* (CMUB39995, holotype). (**a**,**b**) Conidiomata on host substrate (indicated with the red arrow). (**c**,**d**) Section through conidioma. (**e**) Ostiole. (**f**) Conidiomatal walls. (**g**,**h**) Conidiogenous cells giving rise to conidia. (**i**–**k**) Alpha conidia. (**l**,**m**) Colonies on PDA, (**l**) from above and (**m**) from reverse. Scale bars: (**a**,**b**) = 500 μm, (**c**,**d**) = 100 μm, (**e**,**f**) = 20 μm, (**g**,**h**) = 10 μm, and (**i**–**k**) = 5 μm.

***Diaporthe careyae*** Monkai and *S. Lumyong*, sp. nov.

Index Fungorum number: IF900379; Faces of fungi number: FoF 14093; Figure 5

*Etymology*: Refers to the host genus, *Careya*, from which the holotype was collected.

Holotype: CMUB39996

*Saprobic* on dead wood of *Careya sphaerica*. Sexual morph: undetermined. Asexual morph: Coelomycetous. *Conidiomata*: 100–180 high × 150–320 μm diam., pycnidial, immersed to semi-immersed, erumpent, solitary to gregarious, subglobose to ovoid, dark brown to reddish-brown, uni-to bi-loculate, ostiolate, and lacking necks. *Conidiomatal wall*: up to 20 μm wide, comprising a few layers of thick-walled cells, arranged in *textura angularis*, with reddish-brown outer layers and hyaline to brown inner layers. *Conidiophores*: reduced to conidiogenous cells. *Conidiogenous cells*: 4.8–10.7 × 1.4–2.5 μm (*x* = 8 × 2 μm, n = 30), subcylindrical, tapering towards apex, producing 1–2 conidia, hyaline, phialidic, terminal, with visible periclinal thickening and a prominent collarette. *Alpha conidia*: 7–12 × 1.8–3 μm (*x* = 9.4 × 2.6 μm, n = 30), oblong to ellipsoid, obtuse at apex, subtruncate at base, straight to slightly curved or asymmetrical, 0–1(–2) septate, hyaline, smooth-walled, and bi- to multi-guttulate. *Beta conidia*: not observed.

Culture characteristics: *Colonies* on PDA reached 9 cm diam. after 10 days at 25 °C, effuse, sparse hyphae, filiform margin, originally white, becoming grey with age, yellowish to light brown in reverse.

Material examined: Thailand, Chiang Mai Province, Chiang Mai University, near Angkaew Reservoir, on dead wood of *Careya sphaerica* Roxb. (Lecythidaceae), 16 March 2022, J. Monkai, AK02 (CMUB39996, holotype), ex-type living culture, SDBR-CMU469.

Notes: *Diaporthe careyae* formed a well-supported monophyletic lineage basal to species in the *D. carpini* species complex (100% ML, 1.00 PP, Figure 1). Phylogenetically, this species was not clustered with any *Diaporthe* species, and the base pair difference between closely related species was not possible to compare. The morphological characteristics of *D. careyae* are distinct from other *Diaporthe* species in having septate and oblong alpha conidia. Thus, *D. careyae* was proposed as a new species based on its distinctive morphology and phylogenetic placement.

**Figure 5 jof-09-00603-f005:**
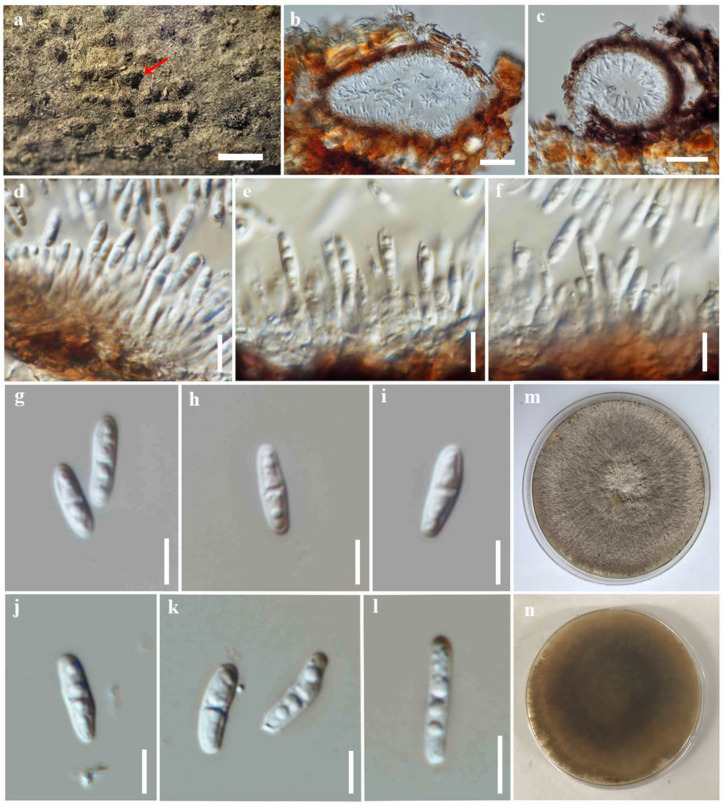
*Diaporthe careyae* (CMUB39996, holotype). (**a**) Conidiomata on host substrate (indicated with the red arrow). (**b**,**c**) Section through conidioma. (**d**–**f**) Conidiogenous cells giving rise to conidia. (**g**–**l**) Alpha conidia. (**m**,**n**) Colonies on PDA, (**m**) from above and (**n**) from reverse. Scale bars: (**a**) = 500 μm, (**b**,**c**) = 50 μm, (**d**–**f**) = 10 μm, and (**g**–**l**) = 5 μm.

***Diaporthe globoostiolata*** Monkai and *S. Lumyong*, sp. nov.

Index Fungorum number: IF900380; Faces of fungi number: FoF 14094; Figure 6

*Etymology*: Refers to the globular shape of the ostiole.

Holotype: MFLU 23-0063

*Saprobic* on dead leaves of a member of Fagaceae. Sexual morph: undetermined. Asexual morph: Coelomycetous. *Conidiomata*: 90–120 high × 110–180 μm diam., pycnidial, semi-immersed, partly erumpent, solitary, subconical to subglobose, dark brown to black, uni-loculate, with ostiolar necks protruding through host surface. *Ostioles*: up to 80 μm wide, central, globose, black, papillate. *Conidiomatal wall*: up to 20 μm wide, comprising a few layers of thin-walled cells, arranged in *textura angularis*, with dark brown outer layers and hyaline to pale brown inner layers. *Conidiophores*: reduced to conidiogenous cells. *Conidiogenous cells*: 3.5–11.4 × 1.4–3.7 μm (*x* = 6.5 × 2.2 μm, n = 30), subcylindrical to ampulliform, slightly tapering towards the apex, hyaline, monophialidic, terminal, with visible periclinal thickening and a prominent collarette. *Alpha conidia*: 6–9.6 × 1.8–2.8 μm (*x* = 7.6 × 2.2 μm, n = 30), fusiform to ellipsoid, obtuse at both ends, aseptate, hyaline, smooth-walled, and mono- to bi-guttulate. *Beta conidia*: 13.2–22 × 1–1.8 μm (*x* = 16.8 ×1.4 μm, n = 30), filiform, tapering towards apex, truncate at base, straight to slightly curved, hyaline, smooth-walled, and eguttulate.

Culture characteristics: *Colonies* on PDA reached 9 cm diam. after 10 days at 25 °C, effuse, fluffy, lobate margin, originally white, becoming pale yellowish mycelium with age, yellowish to pale brown in reverse.

Material examined: Thailand, Tak Province, Tambon Chiang Tong, Wang Chao District, on dead leaves of a member of Fagaceae, 22 August 2019, P. Sysouphanthong, TS1-5 (MFLU 23-0063, holotype), ex-type living culture, MFLUCC 23-0025.

Notes: *Diaporthe globoostiolata* formed a well-supported clade basal to *D. hongkongensis* (99% ML, 1.00 PP, Figure 1). *Diaporthe globoostiolata* can be distinguished from *D. hongkongensis* CBS 115,448 in 1.23%, 3.55%, and 3.67% base pair differences in ITS (7/571 bp), *tef1-α* (12/338 bp), and *tub2* (16/436 bp). *Diaporthe globoostiolata* and *D. hongkongensis* have overlapping sizes of alpha conidia {6–9.6 × 1.8–2.8 vs. (5–)6–7(–8) × (2–)2.5(–3) μm} [15]. However, the beta conidia of *D. globoostiolata* are slightly shorter than those of *D. hongkongensis* {13.2–22 × 1–1.8 vs. 18–22 × 1.5–2 μm} [15]. Our isolate and its closely related taxa, which are *D. hongkongensis* and *D. lithocarpi*, were found on the same host (member of the family, Fagaceae) [30,64,65].

***Diaporthe samaneae*** Monkai and *S. Lumyong*, sp. nov.

Index Fungorum number: IF900381; Faces of fungi number: FoF 14095; Figure 7

*Etymology*: Refers to the host genus, *Samanea*, from which the holotype was collected.

Holotype: CMUB39997

*Saprobic* on dead wood of *Samanea saman*. Sexual morph: undetermined. Asexual morph: Coelomycetous. *Conidiomata*: 300–480 high × 290–740 μm diam., pycnidial, stromatic, superficial to semi-immersed, erumpent, clustered, subglobose to ovoid, elongate, dark brown to brown, multi-loculate, and ostiolate. *Conidiomatal wall*: up to 50 μm wide, comprising a few layers of thin-walled cells, arranged in *textura angularis*, with brown outer layers and hyaline to pale brown inner layers. *Conidiophores*: 7.5–31.7 × 1.5–2.7 μm (*x* = 19 × 2 μm, n = 30), tightly aggregated, subcylindrical, hyaline to pale brown, septate, branched, straight to sinuous, and smooth. *Conidiogenous cells*: 5.2–14.3 × 1.5–2.7 μm (*x* = 9.7 × 2 μm, n = 30), subcylindrical to ampulliform, tapering towards apex, hyaline, phialidic, terminal, with visible periclinal thickening and a prominent collarette. *Alpha conidia*: 7–11 × 1.8–2.8 μm (*x* = 8.4 × 2.4 μm, n = 30), ellipsoid to elongate fusiform, obtuse at apex, subtruncate at base, aseptate, hyaline, smooth-walled, eguttulate, forming basipetal chains of two or more conidia on phialidic neck. *Beta conidia*: not observed.

Culture characteristics: *Colonies* on PDA reached 9 cm diam. after 10 days at 25 °C, effuse, sparse hyphae, filiform margin, originally white, becoming pale yellowish mycelium with age, yellowish to pale brown in reverse, with numerous black dots developing as the fruiting bodies (conidial production not seen).

Material examined: Thailand, Chiang Mai Province, Charoen Prathet Public Park, on dead wood of *Samanea saman* (Jacq.) Merr. (Fabaceae), 27 March 2022, J. Monkai, JS01 (CMUB39997, holotype), ex-type living culture, SDBR-CMU470.

Notes: *Diaporthe samaneae* formed an independent lineage and are closely related to *D. inconspicua* and *D. pseudoinconspicua* (97% ML, 1.00 PP, Figure 1). *Diaporthe samaneae* can be distinguished from *D. inconspicua* CBS 133,813 in 3%, 1.68%, 0%, 2.71%, and 0.84% base pair differences in ITS (17/567 bp), *tef1-α* (5/298 bp), *tub2* (0/423 bp), *cal* (11/406 bp), and *his3* (4/479 bp) and *D. pseudoinconspicua* URM 7874 in 3.93%, 1.08%, 0.64%, 2.47%, and 1.01% base pair differences in ITS (18/458 bp), *tef1-α* (3/278 bp), *tub2* (3/467 bp), *cal* (10/405 bp), and *his3* (5/496 bp). *Diaporthe samaneae* differs from *D. inconspicua* and *D. pseudoinconspicua* in having longer alpha conidia {7–11 × 1.8–2.8 vs. 5.5–6.5 × 1.5–2 μm and 5–7.5(–8.5) ×2–2.5(–3.5) μm} [66,67]. The host preference of *D. inconspicua* is the species of *Maytenus*, *Poincianella*, and *Spondias* [15,66], while *D. pseudoinconspicua* was associated with *Poincianella* [67]. Both species, *D. inconspicua* and *D*. *pseudoinconspicua*, were reported as endophytes, while *D. samaneae* was reported as a saprobe from *Samanea*.

**Figure 6 jof-09-00603-f006:**
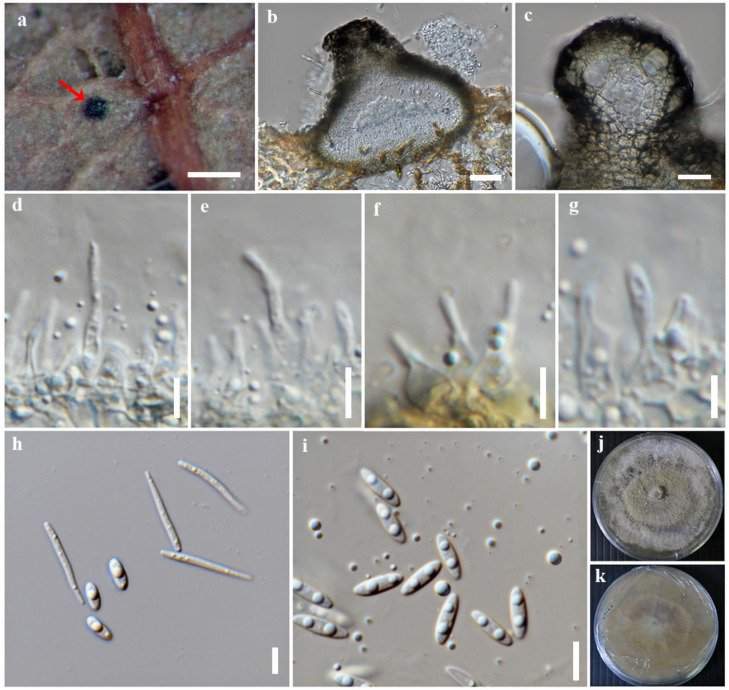
*Diaporthe globoostiolata* (MFLU 23-0063, holotype). (**a**) Conidioma on host substrate (indicated with the red arrow). (**b**) Section through conidioma. (**c**) Ostiole. (**d**–**g**) Conidiogenous cells giving rise to conidia. (**h**,**i**) Alpha and beta conidia. (**j**,**k**) Colonies on PDA, (**j**) from above and (**k**) from reverse. Scale bars: (**a**) = 200 μm, (**b**,**c**) = 20 μm, and (**d**–**i**) = 5 μm.

**Figure 7 jof-09-00603-f007:**
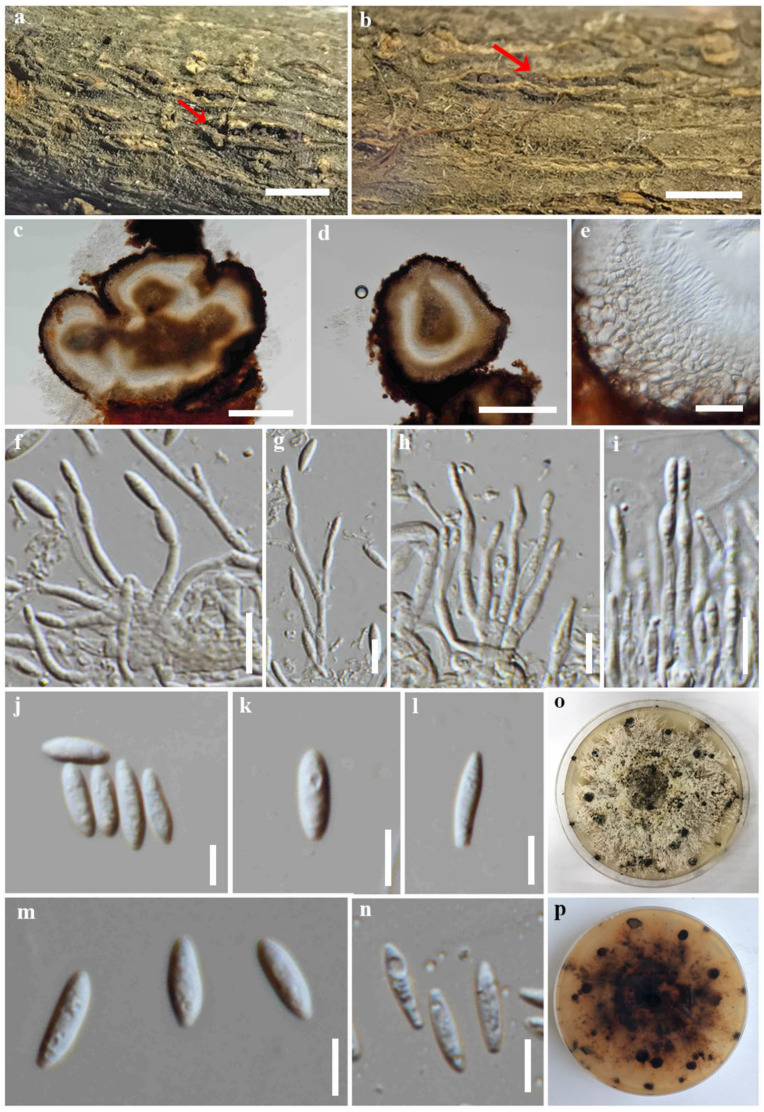
*Diaporthe samaneae* (CMUB39997, holotype). (**a**,**b**) Conidiomata on host substrate (indicated with the red arrow). (**c**,**d**) Section through conidiomata. (**e**) Conidiomatal walls. (**f**–**i**) Conidiogenous cells giving rise to conidia. (**j**–**n**) Alpha conidia. (**o**,**p**) Colonies on PDA, (**o**) from above and (**p**) from reverse. Scale bars: (**a**,**b**) = 500 μm, (**c**,**d**) = 200 μm, (**e**) = 20 μm, (**f**–**i**) = 10 μm, and (**j**–**n**) = 5 μm.

## 4. Discussion

This study describes five novel species of *Diaporthe* in Thailand. Aside from the phenotypic traits, phylogenetic and PHI analyses based on the combined sequence datasets of ITS, *tef1-α*, *tub2*, *cal*, and *his3* were successfully applied to justify the novel species. In particular, *tub2*, *cal*, and *his3* have a high discrimination power for distinguishing species in *Diaporthe*, and this is consistent with the results from other studies [15,18,22,35,36,37].

Our study also gains better insight into the phylogenetic relationships within *Diaporthe*, especially in the *D. arecae* species complex. *Diaporthe zhaoqingensis* and *D. chiangmaiensis* were clustered together in the same clade (98% ML, 1.00 PP) and not so well separated (Figure 1). Therefore, we compared the base pair differences between the type strains of *D. zhaoqingensis* ZHKUCC 22-0056 and *D. chiangmaiensis* MFLUCC 18-0544. There are 1.38% base pair differences in ITS (7/508 bp) between the ex-type of both strains. In the *tef1-α* gene region, there are 0.33% base pair differences (1/300 bp) between the type strains of *D. chiangmaiensis* MFLUCC 18-0544 and *D. zhaoqingensis* ZHKUCC 22-0057. There are 4.94% base pair differences (19/385 bp) in the *tub2* gene region, between *D. chiangmaiensis* MFLUCC 21-0212 and the type strain of *D. zhaoqingensis* ZHKUCC 22-0056. However, some genes from the type strains were not available to compare. The PHI test result also showed that *D. zhaoqingensis* and *D. chiangmaiensis* were not conspecific, indicating that they are different species (Figure 2e). *Diaporthe zhaoqingensis* was isolated as an endophyte on *Morinda officinalis* [18], and *D. chiangmaiensis* was isolated from *Magnolia lilifera* as an endophyte and saprobe [47]. However, the morphological characteristics of these two species could not be compared as only gamma conidia were observed in *D. zhaoqingensis* while alpha conidia were observed in *D. chiangmaiensis* [18,47]. Therefore, more sequence data such as the *tub2*, *cal*, and *his3* of the type strain of *D. chiangmaiensis* are needed to resolve their taxonomic placements and confirm whether they are distinct species.

Furthermore, the new species, *D. careyae*, was shown to be distinct from other *Diaporthe* species based on its morphology and phylogeny. The conidia of *D. careyae* were 0–1(–2) septate, whereas aseptate conidia were a typical characteristic of *Diaporthe*. The septation of conidia has been reported in some *Diaporthe* species (e.g., *D. foeniculina* and *D. saccarata*) [17,68], however, their phylogenetic placements were not closely related to *D. careyae*. It is noteworthy that there are some singleton species that were not grouped into any species complex, and their taxonomic positions remain unclear [32]. In addition, most species of *Diaporthe* lack sequence data and have incomplete morphological descriptions [31,32]; therefore, further extensive sampling is needed in order to unravel the taxonomic circumscription of this genus.

The newly introduced species of *Diaporthe* were associated with different medicinal plants, comprising *D. afzeliae* on *Afzelia xylocarpa*, *D. bombacis* on *Bombax ceiba*, *D. careyae* on *Careya sphaerica*, and *D. samaneae* on *Samanea saman*. These plant species have been used as traditional medicines in tropical countries, including Thailand, and have been reported on concerning their various phytochemicals and pharmacological activities [69,70,71,72,73,74,75]. To the best of our knowledge, none of the *Diaporthe* species have been isolated from these host genera, making this the first report of such a host association [38]. Moreover, a new species, *D. globoostiolata*, was found on a member of Fagaceae. Some plant genera in Fagaceae, such as *Castanopsis*, *Quercus*, and *Lithocarpus*, have also been reported on regarding their medicinal usage and pharmacological properties [76,77,78,79]. Furthermore, more than 30 *Diaporthe* species have been recorded from the host family Fagaceae [38]. This study reflects the high genetic diversity and phenotypic variation within *Diaporthe* and expands our understanding of the diversity and host relationships of the *Diaporthe* species associated with medicinal plants in tropical regions. However, future studies are necessary to investigate the disease symptoms and evaluate the pathogenicity of these *Diaporthe* isolates as they are important for tree health assessments and management.

## Figures and Tables

**Figure 1 jof-09-00603-f001:**
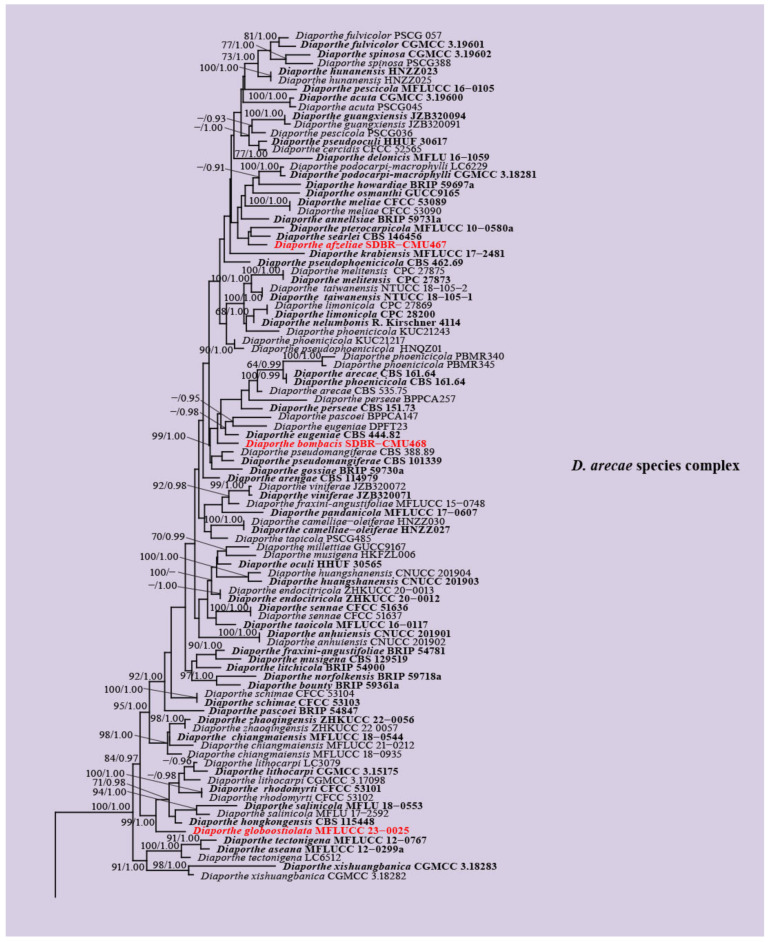
Phylogenetic tree obtained from the RAxML analysis of the combined ITS, *tef1-α*, *tub2*, *cal*, and *his3* sequence data. Bootstrap support values for ML equal to or greater than 60% and Bayesian posterior probabilities equal to or greater than 0.90 PP are indicated at the nodes as ML/PP. The ex-type strains are in black, and the new isolates obtained in this study are in red. The tree is rooted in *Cytospora disciformis* (CBS 116827) and *C. leucostoma* (SXYLt).

**Figure 2 jof-09-00603-f002:**
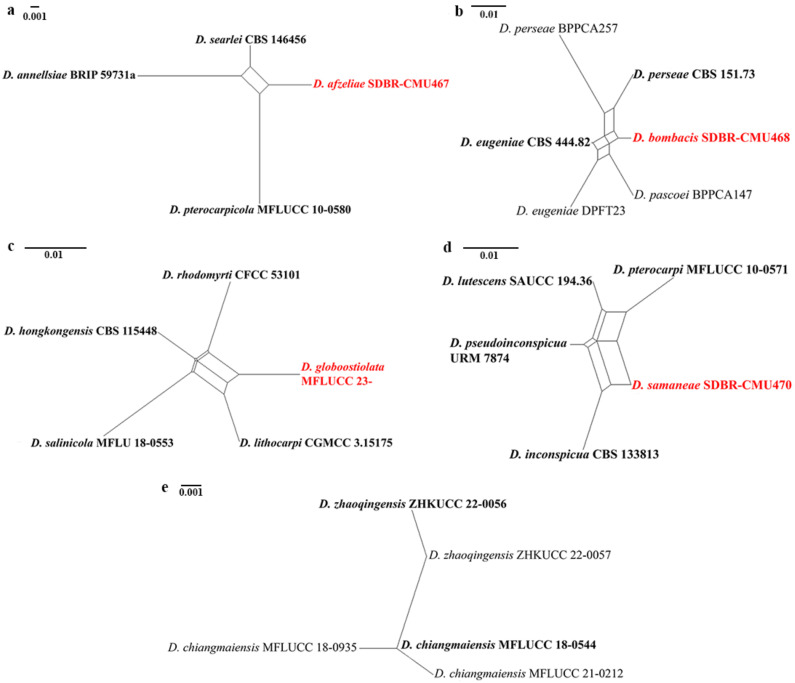
The split graphs of a PHI test result of (**a**) *Diaporthe afzeliae*, (**b**) *D. bombacis*, (**c**) *D. globoostiolata*, and (**d**) *D. samaneae* with their closely related taxa, and (**e**) *D. zhaoqingensis* and *D. chiangmaiensis* using the LogDet transformation and split decomposition options. New species in each graph are indicated in red.

**Table 1 jof-09-00603-t001:** List of taxa and their GenBank accession numbers included in the phylogenetic analyses.

**Taxa Names**	**Culture Accession** **No.**	**GenBank Accession No.**				
		**ITS**	* **tef1-α** *	* **tub2** *	* **cal** *	* **his3** *
*Cytospora disciformis*	CBS 116827 ^T^	KY051801	KX965072	KX964907	N/A	N/A
*C. leucostoma*	SXYLt *	LKEB00000000	N/A	N/A	N/A	N/A
*Diaporthe acuta*	CGMCC 3.19600 ^T^	MK626957	MK654802	MK691225	MK691124	MK726161
*D. acuta*	PSCG045	MK626956	MK654809	MK691223	MK691123	MK726160
** *D. afzeliae* **	**SDBR-CMU467 ^T^**	**OQ600199**	**OQ603502**	**OQ678279**	**OQ646882**	**OQ646886**
*D. ampelina*	CBS 114016 ^T^	AF230751	AY745056	JX275452	AY745026	N/A
*D. anhuiensis*	CNUCC 201901 ^T^	MN219718	MN224668	MN227008	MN224549	MN224556
*D. anhuiensis*	CNUCC 201902	MN219727	MN224669	MN227009	MN224550	MN224557
*D. annellsiae*	BRIP 59731a ^T^	OM918687	OM960596	OM960614	N/A	N/A
*D. arecae*	CBS 161.64 ^T^	KC343032	KC343758	KC344000	KC343274	KC343516
*D. arecae*	CBS 535.75	KC343033	KC343759	KC344001	KC343275	KC343517
*D. arengae*	CBS 114979 ^T^	KC343034	KC343760	KC344002	KC343276	KC343518
*D. aseana*	MFLUCC 12-0299a ^T^	KT459414	KT459448	KT459432	KT459464	N/A
*D. baccae*	CPC 20585	KJ160564	KJ160596	N/A	N/A	N/A
*D. betulicola*	CFCC 51128 ^T^	KX024653	KX024655	KX024657	KX024659	KX024661
*D. betulicola*	CFCC 51129	KX024654	KX024656	KX024658	KX024660	KX024662
*D. bohemiae*	CBS 143347 ^T^	MG281015	MG281536	MG281188	MG281710	MG281361
*D. bohemiae*	CBS 143348	MG281016	MG281537	MG281189	MG281711	MG281362
** *D. bombacis* **	**SDBR-CMU468 ^T^**	**OQ600198**	**OQ603501**	**OQ678278**	**OQ646881**	**OQ646885**
*D. bounty*	BRIP 59361a ^T^	OM918690	OM960599	OM960617	N/A	N/A
*D. camelliae-oleiferae*	HNZZ027 ^T^	MZ509555	MZ504707	MZ504718	MZ504685	MZ504696
*D. camelliae-oleiferae*	HNZZ030	MZ509556	MZ504708	MZ504719	MZ504686	MZ504697
*D. camelliae-sinensis*	SAUCC194.103	MT822631	MT855943	MT855828	MT855710	MT855599
*D. camelliae-sinensis*	SAUCC194.92 ^T^	MT822620	MT855932	MT855817	MT855699	MT855588
*D. canthii*	CBS 132533 ^T^	JX069864	KC843120	KC843230	KC843174	N/A
** *D. careyae* **	**SDBR-CMU469 ^T^**	**OQ600196**	**OQ603449**	**OQ678276**	**OQ646879**	**OQ646883**
*D. carpini*	CBS 114437	KC343044	KC343770	KC344012	KC343286	KC343528
*D. cercidis*	CFCC 52565 ^T^	MH121500	MH121542	MH121582	MH121424	MH121460
*D. chamaeropis*	CBS 454.81 ^T^	KC343048	KC343774	KC344016	KC343290	KC343532
*D. chamaeropis*	CBS 753.70	KC343049	KC343775	KC344017	KC343291	KC343533
*D. chiangmaiensis*	MFLUCC 18-0544 ^T^	OK393703	OL439483	N/A	N/A	N/A
*D. chiangmaiensis*	MFLUCC 18-0935	OK393704	OL439484	N/A	N/A	N/A
*D. chiangmaiensis*	MFLUCC 21-0212	OK393702	OL439482	OK490918	N/A	N/A
*D. cinerascens*	CBS 719.96	KC343050	KC343776	KC344018	KC343292	KC343534
*D. cissampeli*	CBS 141331 ^T^	KX228273	N/A	KX228384	N/A	KX228366
*D. corylicola*	CFCC 53986 ^T^	MW839880	MW815894	MW883977	MW836684	MW836717
*D. cytosporella*	FAU461 ^T^	KC843307	KC843116	KC843221	KC843141	MF418283
*D. decedens*	CBS 109772	KC343059	KC343785	KC344027	KC343301	KC343543
*D. decedens*	CBS 114281	KC343060	KC343786	KC344028	KC343302	KC343544
**Taxa Names**	**Culture Accession** **No.**	**GenBank Accession No.**				
		**ITS**	** *tef1-α* **	** *tub2* **	** *cal* **	** *his3* **
*D. decorticans*	CBS 114200	KC343169	KC343895	KC344137	KC343411	KC343653
*D. decorticans*	CBS 114649	KC343170	KC343896	KC344138	KC343412	KC343654
*D. delonicis*	MFLU 16-1059 ^T^	MT215490	N/A	MT212209	N/A	N/A
*D. detrusa*	CBS 109770	KC343061	KC343787	KC344029	KC343303	KC343545
*D. detrusa*	CBS 114652	KC343062	KC343788	KC344030	KC343304	KC343546
*D. elaeagni-glabrae*	CGMCC 3.18287 ^T^	KX986779	KX999171	KX999212	KX999281	KX999251
*D. elaeagni-glabrae*	LC4806	KX986780	KX999172	KX999213	KX999282	KX999252
*D. endocitricola*	ZHKUCC 20-0012 ^T^	MT355682	MT409336	MT409290	MT409312	N/A
*D. endocitricola*	ZHKUCC 20-0013	MT355683	MT409337	MT409291	MT409313	N/A
*D. eugeniae*	CBS 444.82 ^T^	KC343098	KC343824	KC344066	KC343340	KC343582
*D. eugeniae*	DPFT23	MK110366	MK117267	MK122799	N/A	N/A
*D. fibrosa*	CBS 109751	KC343099	KC343825	KC344067	KC343341	KC343583
*D. fibrosa*	CBS 113830	KC343100	KC343826	KC344068	KC343342	KC343584
*D. foeniculina*	CBS 111553 ^T^	KC843295	KC843104	KC843209	KC843129	N/A
*D. foeniculina (=D. foeniculacea)*	CBS 123208 ^T^	KC343104	KC343830	KC344072	KC343346	KC343588
*D. foeniculina (=D. rhoicola)*	CBS 129528 ^T^	JF951146	KC843100	KC843205	KC843124	N/A
*D. forlicesenica*	MFLUCC 17-1015 ^T^	KY964215	KY964171	KY964099	N/A	N/A
*D. fraxini-angustifoliae*	BRIP 54781 ^T^	JX862528	JX862534	KF170920	N/A	N/A
*D. fraxini-angustifoliae*	MFLUCC 15-0748	KT459428	KT459446	KT459430	KT459462	N/A
*D. fulvicolor*	CGMCC 3.19601 ^T^	MK626859	MK654806	MK691236	MK691132	MK726163
*D. fulvicolor*	PSCG 057	MK626858	MK654810	MK691233	MK691131	MK726164
** *D. globoostiolata* **	**MFLUCC 23-0025 ^T^**	**OQ600200**	**OQ603503**	**OQ678280**	**N/A**	**N/A**
*D. gossiae*	BRIP 59730a ^T^	OM918693	OM960602	OM960620	N/A	N/A
*D. guangxiensis*	JZB320091	MK335769	MK523564	MK500165	MK736724	N/A
*D. guangxiensis*	JZB320094 ^T^	MK335772	MK523566	MK500168	MK736727	N/A
*D. hickoriae*	CBS 145.26 ^T^	KC343118	KC343844	KC344086	KC343360	KC343602
*D. hispaniae*	CBS 143351 ^T^	MG281123	MG281644	MG281296	MG281820	MG281471
*D. hispaniae*	CBS 143352	MG281124	MG281645	MG281297	MG281821	MG281472
*D. hongkongensis*	CBS 115448 ^T^	KC343119	KC343845	KC344087	KC343361	KC343603
*D. howardiae*	BRIP 59697a ^T^	OM918695	OM960604	OM960622	N/A	N/A
*D. huangshanensis*	CNUCC 201903 ^T^	MN219729	MN224670	MN227010	N/A	MN224558
*D. huangshanensis*	CNUCC 201904	MN219730	MN224671	MN227011	N/A	MN224559
*D. hunanensis*	HNZZ023 ^T^	MZ509550	MZ504702	MZ504713	MZ504680	MZ504691
*D. hunanensis*	HNZZ025	MZ509551	MZ504703	MZ504714	MZ504681	MZ504692
*D. hungariae*	CBS 143353 ^T^	MG281126	MG281647	MG281299	MG281823	MG281474
*D. hungariae*	CBS 143354	MG281127	MG281648	MG281300	MG281824	MG281475
**Taxa Names**	**Culture Accession** **No.**	**GenBank Accession No.**				
		**ITS**	** *tef1-α* **	** *tub2* **	** *cal* **	** *his3* **
*D. impulsa*	CBS 114434	KC343121	KC343847	KC344089	KC343363	KC343605
*D. impulsa*	CBS 141.27	KC343122	KC343848	KC344090	KC343364	KC343606
*D. inconspicua*	CBS 133813 ^T^	KC343123	KC343849	KC344091	KC343365	KC343607
*D. inconspicua*	URM7776	MG696772	MG710414	MG710395	MG710391	MG710410
*D. isoberliniae*	CPC 22549 ^T^	KJ869133	N/A	KJ869245	N/A	N/A
*D. juglandicola*	CFCC 51134 ^T^	KU985101	KX024628	KX024634	KX024616	KX024622
*D. juglandicola*	CFCC 51135	KU985102	KX024629	KX024635	KX024617	KX024623
*D. krabiensis*	MFLUCC 17-2481 ^T^	MN047101	MN433215	MN431495	N/A	N/A
*D. limonicola*	CPC 27869	MF418419	MF418498	MF418579	MF418253	MF418339
*D. imonicola*	CPC 28200 ^T^	NR_154980	MF418501	MF418582	MF418256	MF418342
*D. litchiicola*	BRIP 54900 ^T^	JX862533	JX862539	KF170925	N/A	N/A
*D. lithocarpi*	CGMCC 3.15175 ^T^	KC153104	KC153095	KF576311	KF576236	N/A
*D. ithocarpi*	CGMCC 3.17098	KF576276	KF576251	KF576300	KF576228	N/A
*D. lithocarpi*	LC3079	KP267851	KP267925	KP293431	N/A	KP293505
*D. lutescens*	SAUCC 194.36 ^T^	MT822564	MT855877	MT855761	MT855647	MT855533
*D. macintoshii*	BRIP 55064a ^T^	KJ197289	KJ197251	KJ197269	N/A	N/A
*D. maytenicola*	CPC 21896 ^T^	KF777157	N/A	KF777250	N/A	N/A
*D. melastomatis*	SAUCC194.55 ^T^	MT822583	MT855896	MT855780	MT855664	MT855551
*D. melastomatis*	SAUCC194.80	MT822608	MT855920	MT855805	MT855687	MT855576
*D. meliae*	CFCC 53089 ^T^	MK432657	ON081654	MK578057	N/A	ON081662
*D. meliae*	CFCC 53090	MK432658	ON081655	MK578058	N/A	ON081663
*D. melitensis*	CPC 27873 ^T^	MF418424	MF418503	MF418584	MF418258	MF418344
*D. melitensis*	CPC 27875	MF418425	MF418504	MF418585	MF418259	MF418345
*D. millettiae*	GUCC9167 ^T^	MK398674	MK480609	MK502089	MK502086	N/A
*D. musigena*	CBS 129519 ^T^	KC343143	KC343869	KC344111	KC343385	KC343627
*D. musigena*	HKFZL006	MK050110	MK054238	MK079660	N/A	N/A
*D. nebulae*	Phom240	KY511315	MH708543	KY511346	N/A	N/A
*D. nebulae*	PMM1681 ^T^	KY511337	MH708552	KY511369	N/A	N/A
*D. nelumbonis*	R. Kirschner 4114 ^T^	KT821501	N/A	LC086652	N/A	N/A
*D. nigra*	JZB320170 ^T^	MN653009	MN892277	MN887113	N/A	N/A
*D. norfolkensis*	BRIP 59718a ^T^	OM918699	OM960608	OM960626	N/A	N/A
*D. oculi*	HHUF 30565 ^T^	LC373514	LC373516	LC373518	N/A	N/A
*D. oncostoma*	CBS 100454	KC343160	KC343886	KC344128	KC343402	KC343644
*D. oncostoma*	CBS 589.78	KC343162	KC343888	KC344130	KC343404	KC343646
*D. osmanthi*	GUCC9165 ^T^	MK398675	MK480610	MK502091	MK502087	N/A
*D. pandanicola*	MFLUCC 17-0607 ^T^	MG646974	N/A	MG646930	N/A	N/A
*D. parapterocarpi*	CPC 22729 ^T^	KJ869138	N/A	KJ869248	N/A	N/A
*D. parvae*	CGMCC 3.19599 ^T^	MK626919	MK654858	MK691248	N/A	MK726210
*D. parvae*	PSCG035	MK626920	MK654859	MK691249	MK691169	MK726211
*D. pascoei*	BPPCA147	MK111091	MK117255	MK122790	N/A	N/A
**Taxa Names**	**Culture Accession** **No.**	**GenBank Accession No.**				
		**ITS**	** *tef1-α* **	** *tub2* **	** *cal* **	** *his3* **
*D. pascoei*	BRIP 54847 ^T^	JX862532	JX862538	KF170924	N/A	N/A
*D. perseae*	BPPCA257	MK111098	MK117256	MK122791	N/A	N/A
*D. perseae*	CBS 151.73 ^T^	KC343173	KC343899	KC344141	KC343415	KC343657
*D. pescicola*	MFLUCC 16-0105 ^T^	KU557555	KU557623	KU557579	KU557603	N/A
*D. pescicola*	PSCG036	MK626855	MK654796	MK691226	MK691116	MK726159
*D. phillipsii*	CAA817 ^T^	MK792305	MK828076	MN000351	MK883831	MK871445
*D. phillipsii*	CAA818	MK792307	MK828078	MN000352	MK883833	MK871447
*D. phoenicicola*	CBS 161.64 ^T^	MH858400	GQ250349	JX275440	JX197432	N/A
*D. phoenicicola*	KUC21217	KT207733	N/A	KT207633	N/A	N/A
*D. phoenicicola*	KUC21243	KT207761	N/A	KT207659	N/A	N/A
*D. phoenicicola*	PBMR340	MK111086	MK117271	MK122805	N/A	N/A
*D. phoenicicola*	PBMR345	MK111088	MK117275	MK122810	N/A	N/A
*D. podocarpi-macrophylli*	CGMCC 3.18281 ^T^	KX986774	KX999167	KX999207	KX999278	KX999246
*D. podocarpi-macrophylli*	LC6229	KX986771	KX999164	KX999204	KX999277	KX999243
*D. poincianellae*	URM 7932 ^T^	MH989509	MH989538	MH989537	MH989540	MH989539
*D. portugallica*	CPC 34247 ^T^	MH063905	MH063911	MH063917	MH063893	MH063899
*D. portugallica*	CPC 34248	MH063906	MH063912	MH063918	MH063894	MH063900
*D. pseudoinconspicua*	URM 7873	MH122535	MH122530	MH122521	MH122525	MH122518
*D. pseudoinconspicua*	URM 7874 ^T^	MH122538	MH122533	MH122524	MH122528	MH122517
*D. pseudomangiferae*	CBS 101339 ^T^	KC343181	KC343907	KC344149	KC343423	KC343665
*D. pseudomangiferae*	CBS 388.89	KC343182	KC343908	KC344150	KC343424	KC343666
*D. pseudooculi*	HHUF 30617 ^T^	NR_161019	LC373517	LC373519	N/A	N/A
*D. pseudophoenicicola*	CBS 462 69 ^T^	KC343184	KC343910	KC344152	KC343426	KC343668
*D. pseudophoenicicola*	HNQZ01	MN424520	MN424562	MN424534	MN424576	MN424548
*D. psoraleae*	CBS 136412 ^T^	KF777158	KF777245	KF777251	N/A	N/A
*D. psoraleae pinnatae*	CBS 136413 ^T^	KF777159	N/A	KF777252	N/A	N/A
*D. pterocarpi*	MFLUCC 10-0571 ^T^	JQ619899	JX275416	JX275460	JX197451	N/A
*D. pterocarpi*	MFLUCC 10-0588	JQ619900	JX275417	JX275461	JX197452	N/A
*D. pterocarpicola*	MFLUCC 10-0580 ^T^	JQ619887	JX275403	JX275441	JX197433	N/A
*D. pungensis*	SAUCC 194.112 ^T^	MT822640	MT855952	MT855837	MT855719	MT855607
*D. pungensis*	SAUCC 194.89	MT822617	MT855929	MT855814	MT855696	MT855585
*D. ravennica*	MFLUCC 15-0479 ^T^	KU900335	N/A	KX432254	N/A	N/A
*D. ravennica*	MFLUCC 17-1029	KY964191	KY964147	KY964075	N/A	N/A
*D. rhodomyrti*	CFCC 53101 ^T^	MK432643	MK578119	MK578046	MK442965	MK442990
**Taxa Names**	**Culture Accession** **No.**	**GenBank Accession No.**				
		**ITS**	** *tef1-α* **	** *tub2* **	** *cal* **	** *his3* **
*D. rhodomyrti*	CFCC 53102	MK432644	MK578120	MK578047	MK442966	MK442991
*D. rostrata*	CFCC 50062 ^T^	KP208847	KP208853	KP208855	KP208849	KP208851
*D. rostrata*	CFCC 50063	KP208848	KP208854	KP208856	KP208850	KP208852
*D. rumicicola*	JZB320006 ^T^	MK066126	MK078545	MK078546	N/A	N/A
*D. rumicicola*	MFLUCC 18-0739	MH846233	MK049554	MK049555	N/A	N/A
*D. saccarata*	CBS 116311 ^T^	KC343190	KC343916	KC344158	KC343432	KC343674
*D. salinicola*	MFLU 17-2592	MN047099	MN077074	N/A	N/A	N/A
*D. salinicola*	MFLU 18-0553 ^T^	MN047098	MN077073	N/A	N/A	N/A
** *D. samaneae* **	**SDBR-CMU470 ^T^**	**OQ600197**	**OQ603500**	**OQ678277**	**OQ646880**	**OQ646884**
*D. schimae*	CFCC 53103 ^T^	MK432640	MK578116	MK578043	MK442962	MK442987
*D. schimae*	CFCC 53104	MK432641	MK578117	MK578044	MK442963	MK442988
*D. schisandrae*	CFCC 51988 ^T^	KY852497	KY852509	KY852513	KY852501	KY852505
*D. schisandrae*	CFCC 51989	KY852498	KY852510	KY852514	KY852502	KY852506
*D. scobina*	CBS 251.38	KC343195	KC343921	KC344163	KC343437	KC343679
*D. searlei*	CBS 146456 ^T^	MN708231	N/A	MN696540	N/A	N/A
*D. sennae*	CFCC 51636 ^T^	KY203724	KY228885	KY228891	KY228875	N/A
*D. sennae*	CFCC 51637	KY203725	KY228886	KY228892	KY228876	N/A
*D. spinosa*	CGMCC 3.19602 ^T^	MK626849	MK654811	MK691234	MK691129	MK726156
*D. spinosa*	PSCG388	MK626860	MK654798	MK691229	MK691128	MK726171
*D. stictica*	CBS 370.54 ^T^	KC343212	KC343938	KC344180	KC343454	KC343696
*D. taiwanensis*	NTUCC 18-105-1 ^T^	MT241257	MT251199	MT251202	MT251196	N/A
*D. taiwanensis*	NTUCC 18-105-2	MT241258	MT251200	MT251203	MT251197	N/A
*D. taoicola*	MFLUCC 16-0117 ^T^	KU557567	KU557635	KU557591	N/A	N/A
*D. taoicola*	PSCG485	MK626869	MK654812	MK691227	MK691120	MK726173
*D. tectonigena*	LC6512	KX986782	KX999174	KX999215	KX999284	KX999254
*D. tectonigena*	MFLUCC 12-0767 ^T^	KU712429	KU749371	KU743976	KU749358	N/A
*D. thunbergiae*	MFLUCC 10-0576a ^T^	JQ619893	JX275409	JX275449	JX197440	N/A
*D. thunbergiae*	MFLUCC 10-0576b	JQ619894	JX275410	JX275450	JX197441	N/A
*D. toxicodendri*	FFPRI420990	LC275193	LC275217	LC275225	LC275201	LC275209
*D. vangueriae*	CPC 22703 ^T^	KJ869137	N/A	KJ869247	N/A	N/A
*D. velutina*	CGMCC 3.18286 ^T^	KX986790	KX999182	KX999223	N/A	KX999261
*D. velutina*	LC4419	KX986789	KX999181	KX999222	KX999286	KX999260
*D. verniciicola*	CFCC 53109 ^T^	MK573944	MK574619	MK574639	MK574583	MK574599
*D. verniciicola*	CFCC 53110	MK573945	MK574620	MK574640	MK574584	MK574600
*D. viniferae*	JZB320071 ^T^	MK341550	MK500107	MK500112	MK500119	N/A
*D. viniferae*	JZB320072	MK341551	MK500108	MK500113	MK500120	N/A
*D. woolworthii*	CBS 148.27	KC343245	KC343971	KC344213	KC343487	KC343729
*D. xishuangbanica*	CGMCC 3.18282	KX986783	KX999175	KX999216	N/A	KX999255
*D. xishuangbanica*	CGMCC 3.18283 ^T^	KX986784	KX999176	KX999217	N/A	N/A
*D. zaobaisu*	CGMCC 3.19598 ^T^	MK626922	MK654855	MK691245	N/A	MK726207
**Taxa Names**	**Culture Accession** **No.**	**GenBank Accession No.**				
		**ITS**	** *tef1-α* **	** *tub2* **	** *cal* **	** *his3* **
*D. zaobaisu*	PSCG032	MK626923	MK654856	MK691246	N/A	MK726208
*D. zhaoqingensis*	ZHKUCC 22-0056 ^T^	ON322885	N/A	ON315074	ON315000	ON315015
*D. zhaoqingensis*	ZHKUCC 22-0057	ON322886	ON315043	ON315075	N/A	ON315016

The ex-type cultures are indicated with the superscript “^T^” and the newly generated sequences are indicated in bold. “N/A” indicates the sequence is unavailable. “*” indicates a whole genomic DNA strain.

## Data Availability

All sequences generated in this study were submitted to GenBank (https://www.ncbi.nlm.nih.gov, accessed on 1 April 2023).

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
