# Peer review of "Integrative Taxonomy of Novel Diaporthe Species Associated with Medicinal Plants in Thailand"

_jof, 2023, doi:10.3390/jof9060603_

Round 1

Reviewer 1 Report

This is a well-orgnized and written paper. Five new species were described and illustrated based on a polyphasic approch. The descriptions are standard and the illustrations are in good condition. I have only two small suggestions:

1. The two species D. zhaoqingensis and D. chiangmaiensis were studied in the paper. This content should be included in the abstract section.

2. It is good for readers who don't work on the genus that the authors can add some arrows on the photos of the hosts to show the conidiomata.

Author Response

Thanks for your comments and suggestions. We have revised the paper accordingly. Please kindly check the responses as follow:

Point 1: The two species D. zhaoqingensis and D. chiangmaiensis were studied in the paper. This content should be included in the abstract section.

Response 1: We added the sentence “Our phylogeny also reveals the close relationship between D. zhaoqingensis and D. chiangmaiensis; however, the evidences of PHI test and DNA comparison support that they are distinct species.” in the abstract (line 21–23).

Point 2: It is good for readers who don't work on the genus that the authors can add some arrows on the photos of the hosts to show the conidiomata.

Response 2: We added the arrows to point out the conidiomata on host substrate in the Figure 3–7.

Reviewer 2 Report

The current study introduces Diaporthe afzeliae, D. bombacis, D. careyae, D. globoostiolata and D. samaneae as new species, using the phenotypic traits, phylogenetic and PHI analyses. Diaporthe species are a large and diverse genus whose members are mostly known as endophytes, saprobes and plant pathogens. Interestingly, this study is the first report of Diaporthe species associated with medicinal plants in tropical regions. Therefore, the discovery of new species within the genus are really important to help researchers better understand the diversity and host association with medicinal plants. The paper is well presented with numerous references to previous works. Good writing, good results and good discussion. To conclude my review, I am impressed with paper and is worth being published after minor corrections. I have included my comments in the attached PDF.

The overall quality of English language is good!

Author Response

Thanks for your comments and suggestions. We revised the paper following your comments, the changes were marked up with yellow highlight.  Please kindly check in the revised version. 

Reviewer 3 Report

It is surprising that an indeterminate species, whatever the family, be used as a medicinal plant. I refer to the host of Diaporthe globoostiolata. Could the authors make an effort to identify the plant?

In Figs. 3a, 3b, 4a, 4b, 5a, 6a, 7a, 7b, the scale bar is missing.

The characteristics of the culture are given, I understand, after 10 days. However, the pictures seem to have not been taken after 10 days. For example, figures 3a, 3b and 4l, 4m seem to show two colonies that cannot reach 9 mm as stated in the text. Figures 5m and 5n show a full occupation of the Petri dish, while the text says that the colony reaches 5 cm in 10 days.
